# Photoluminescence Characteristics of Sn^2+^ and Ce^3+^-Doped Cs_2_SnCl_6_ Double-Perovskite Crystals

**DOI:** 10.3390/ma12091501

**Published:** 2019-05-08

**Authors:** Hongdan Zhang, Ludan Zhu, Jun Cheng, Long Chen, Chuanqi Liu, Shuanglong Yuan

**Affiliations:** Key Laboratory for Ultrafine Materials of Ministry of Education, School of Materials Science and Engineering, East China University of Science and Technology, Shanghai 200237, China; Y30160497@mail.ecust.edu.cn (H.Z.); Y30170516@mail.ecust.edu.cn (L.Z.); Y30170392@mail.ecust.edu.cn (J.C.); Y30180381@mail.ecust.edu.cn (L.C.); Y30180431@mail.ecust.edu.cn (C.L.)

**Keywords:** lead-free perovskite, Cs_2_SnCl_6_, doping, photoluminescence

## Abstract

In recent years, all-inorganic lead-halide perovskites have received extensive attention due to their many advantages, but their poor stability and high toxicity are two major problems. In this paper, a low toxicity and stable Cs_2_SnCl_6_ double perovskite crystals were prepared by aqueous phase precipitation method using SnCl_2_ as precursor. By the XRD, ICP-AES, XPS, photoluminescence and absorption spectra, the fluorescence decay curve, the structure and photoluminescence characteristics of Ce^3+^-doped and undoped samples have been investigated in detail. The results show that the photoluminescence originates from defects. [$Sn_{Sn^{4+}}^{2+}$+V_Cl_] defect complex in the crystal is formed by Sn^2+^ substituting Sn^4+^. The number of defects formed by Sn^2+^ in the crystal decreases with Ce^3+^ content increases. Within a certain number of defects, the crystal luminescence is enhanced with the number of [$Sn_{Sn^{4+}}^{2+}$+V_Cl_] decreased. When Ce^3+^ is incorporated into the crystals, the defects of [$Ce^{3+}{}_{Sn^{4+}}$+V_Cl_] and [$Sn_{Sn^{4+}}^{2+}$+V_Cl_] were formed and the crystal show the strongest emission. This provides a route to enhance the photoluminescence of Cs_2_SnCl_6_ double perovskite crystals.

## 1. Introduction

In recent years, the excellent performance of lead halide perovskite in optical instruments has attracted increasing attention in the professional sphere [1]. Since the Mitzi research group [2] first reported organic-inorganic hybrid perovskite material and found its strong electron migration capability at the end of the 20th century. The great potential of applying its capability to solar cells has inspired a new wave of research at the time [3,4,5]. Using the liquid phase method, the researchers synthesized all inorganic lead-halide perovskite CsPbX_3_ (X = Cl,Br,I) nanocrystal [6,7], which has the same structure as hybrid perovskite [8]. The synthesized nanocrystals have captured much attention with their superior features, including high fluorescence quantum efficiency (up to 100%) [9], light emission wavelength covering the entire visible spectrum (400–700 nm) [10], relatively narrow full width at half maximum (FWHM) (12–42 nm) [11] etc. Therefore, they have shown a great application potential in the light emitting diode (LED) [12,13], quantum dot light emitting diodes (QLED) [14], laser [15,16], fluorescent probe [17] and the energy conversion efficiency of CH_3_NH_3_PbI_3_ and CsPbI_3_ solar cells has reached 23% [18]. Although CH_3_NH_3_PbX_3_ and CsPbX_3_ exhibit excellent performance, there still remains a major concern on its application, namely the high toxicity of lead [19]. If the lead is always included in the solar photovoltaic module, it will not cause any problems. However, the fact that lead-based perovskite would normally release PbX_2_ as a degradation product, which would cause negative effects on health and the environment [20]. The global demand for renewable and green energy has triggered the exploration of low-cost, high-efficiency photovoltaic device. In order to realize a wider application of perovskite device, replacing the unsafe lead component seems to be vital. The tin-based perovskite CsSnX_3_ is a very good substitute for applications in solar cells, infrared light-emitting diodes and lasers [21,22,23,24,25]. Unfortunately, the Sn^2+^ in CsSnX_3_ perovskites is easily oxidized to Sn^4+^, resulting in high sensitivity to ambient atmospheres (oxygen, moisture, etc.) [21,22,26,27].

As a defect variant of the traditional cubic ABX_3_ perovskite, Cs_2_SnX_6_ has a crystal structure similar to that of CsPbX_3_. Cs_2_SnX_6_ is expected to be applied to large-scale production [28] since it has considerable stability to oxygen and moisture. The high stability is attributed to the high value Sn^4+^ and higher decomposition enthalpy (1.37 eV per formula unit for Cs_2_SnCl_6_) than that of traditional ABX_3_ halide perovskites (e.g., 0.29 eV for CsPbBr_3_) [29]. Many publications have reported recently the preparation method and potential applications of Cs_2_SnX_6_ perovskite crystals. Among those, Wang A et al. [1] synthesized Cs_2_SnI_6_ nanocrystals and proved that the Cs_2_SnX_6_ based field effect transistor has the characteristics of a P-type semiconductor with high hole mobility (>20 cm^2^/(Vs)). Kaltzoglou A et al. synthesized Cs_2_SnCl_6_,Cs_2_SnBr_6_,Cs_2_SnI_6_ and Cs_2_SnI_3_Br_3_ crystals under different temperatures using different solvents [28,30]. These crystals were used as hole transport materials in dye-sensitized solar cells, its power conversion efficiency has reached maximum 4.23% under one solar radiation. At meanwhile, Saparov B et al. [19] obtained an n-type Cs_2_SnI_6_ semiconductor film by annealing SnI_4_ vapor onto a glass substrate covered with a CsI film. Xiaofeng Qiu et al. [22] oxidized CsSnI_3_ film into Cs_2_SnI_6_ film as the light absorbing layer in solar cells.

Nevertheless, there are very few reports on the luminescence properties of the Cs_2_SnCl_6_ crystal, except the recent one about strong blue light emitting by doping Bi^3+^ [29]. Actually, a few Sn^2+^ ions play a key role in producing Bi_Sn_+V_Cl_ defect complex, which is responsible for the strong blue emission according to the literature [29]. It is unclear whether Sn^2+^ itself or the other trivalent ions have similar behavior. Therefore, in this paper, SnCl_2_ and CsCl are used as precursors to synthesize Cs_2_SnCl_6_ crystals by liquid phase precipitation method. The blue light-emitting crystals in which defect band reduce the original band gap in the forbidden band were obtained by doping and the effects of Sn^2+^ and Ce^3+^ on the luminescence properties of the crystal have been investigated.

## 2. Materials and Methods

### 2.1. Chemicals

Tin (Sn, 99.9%, Aladdin, Shanghai, China), cesium chloride (CsCl, 99.99%, Aladdin, Shanghai, China), cerium chloride heptahydrate (CeCl_3_·7H_2_O, 99.99%, Henghua, Ninan, China), hydrochloric acid (HCl, 36–38%, Titan, Shanghai, China), deionized water (H_2_O), absolute alcohol (C_2_H_6_O, Titan, Shanghai, China).

### 2.2. Preparation of SnCl_2_ Solution

The Sn powder and HCl were taken in a beaker at a molar ratio of 1:4 and the beaker was heated at 90 °C for ensuring complete dissolution of tin powder to prepare SnCl_2_ solution. Since Sn^2+^ easily was oxidized to Sn^4+^ under certain conditions, leading to the solution containing SnCl_2_ and SnCl_4_ (for convenience, it was also represented by SnCl_2_ hereinafter).

### 2.3. Synthesis of Cs_2_SnCl_6_ Crystals

The beaker containing 0.036 mol CsCl, 60 mL concentrated hydrochloric acid and 78 mL deionized water was heated to 90 °C in a water bath. 0.018 mol of newly synthesized SnCl_2_ was injected by a pipette with stirring. 1/6 samples were taken out at different time point (0 min, 10 min, 60 min, 360 min, 660 min, 1020 min). The samples were naturally cooled to room temperature and centrifuged to obtain a precipitate, which was washed twice with absolute ethanol to remove hydrochloric acid on the surface of the samples and naturally dried in the air. The samples have been numbered as A1–A6 in Table 1.

### 2.4. Synthesis of Ce^3+^-Doped Cs_2_SnCl_6_ Crystals

Firstly, the beaker containing a certain amount of CsCl, different amount of CeCl_3_·7H_2_O, 10 mL concentrated hydrochloric acid and 13 mL deionized water was heated to 90 °C and then 0.003 mol of SnCl_2_ was injected by a pipette. Secondly, the mixture has reacted for 11 h at 90 °C and then naturally cooled to room temperature and centrifuged to obtain precipitate. Finally, the samples were washed twice with absolute ethanol to remove hydrochloric acid and Ce^3+^ on the surface of the samples and naturally dried in the air. The samples have been numbered as C1–C5 in Table 1.

The same preparation steps have been employed to investigate the effect of different reaction time for Ce^3+^-doped Cs_2_SnCl_6_ samples by adding 3-fold amount of CsCl, concentrated hydrochloric acid, deionized water and SnCl_2_ to the beaker, 1/3 volume of solution was taken for measurements. The samples have been numbered as B1–B3 in Table 1.

### 2.5. Measurements

The chemical states of Cs, Sn, Ce and Cl were determined by X-ray photoelectron spectroscopy (XPS, ESCALAB 250Xi, Thermo Fisher, Basingstoke, UK). The XRD patterns of the sample were measured by an X-ray diffractometer (D/MAX 2550 VB/PC, Rigaku Corporation, Tokyo, Japan). The doping concentration of Ce^3+^ was determined by inductively coupled plasma optical emission spectrometry (ICP-AES, Agilent, Santa Clara, CA, USA). The reflectance and fluorescence spectra of the samples were measured by an UV-visible spectrophotometer (UV-2550, Shimadzu, Kyoto, Japan) and a molecular fluorescence spectrometer (Fluorolog-3-P, Jobin Yvon, Paris, France). The fluorescence quantum efficiency and fluorescence lifetime of the Cs_2_SnCl_6_: Ce were determined by a fluorescence spectrometer with an integrating sphere (FLS980, Edinburgh instrument Ltd., Livingston, UK).

## 3. Results and Discussion

The absorption spectra of 25% Ce^3+^-doped Cs_2_SnCl_6_ crystals at different reaction time are shown in Figure 1. All samples have shown a same strong absorption band with an edge (Figure 1b) at 312 nm (3.97 eV), which is consistent with the absorption edge reported in the literature at 317 nm (3.9 eV) [28] and 313 nm (3.96 eV) [31], indicating this absorption band originates from the absorption of the Cs_2_SnCl_6_ matrix. In addition to the absorption of the matrix, the absorption spectra have exhibited an extra weak absorption band between 312 and 390 nm. The similar weak absorption band has also been exhibited when doping Bi^3+^ in the literature [29]. In this literature, it figured out that the formation of [V_Cl_+Bi_Sn_] defect complex by introducing Bi^3+^ creates a defect energy band composed of Cl 3p and Bi 6s orbitals above the valence band maximum (VBM), resulting in the additional absorption band. Therefore, it is reasonable to infer that the extra weak absorption band in the absorption spectra in this work is similar to the literature [29], but which might be caused by the defect formed by Ce^3+^ or Sn^2+^ doping because of the absence of Bi^3+^. the absorption intensity increases as the number of defects increases. The intensities of the weak absorption bands increase in the order of B2, B3 and B1, which means contrary defect concentration order due to the positive correlation between the absorption intensity and the defect concentration. 

To further demonstrate the above inference, the measured ICP-AES data of the samples were also listed in Table 1. Only the B2 sample contained a few of Ce^3+^ with a mass of 0.02% among three samples. Figure 2 has shown the XPS spectra of B1 and B2 and the fitted spectra of high-resolution Sn 3d. Only the energy peaks of Cs, Sn and Cl appeared in the total spectra. The Sn 3d spectra composed of two asymmetrical peaks at about 496 eV and 487 eV respectively. To clarify the origin of two peaks, they were fitted by using Sn^4+^ (Figure 2c,e) and Sn^4+^/Sn^2+^ pair (Figure 2d,f), as a result the later exhibited better fitting degree. It suggests the existence of the characteristics 3d_3/2_ (495.7 eV) and 3d_5/2_ (487.2 eV) peaks of Sn^2+^ in addition to Sn^4+^ 3d_3/2_ (496.1 eV) and 3d_5/2_ (487.6 eV) [29,32]. Furthermore, comparing the Ce 3d high-resolution spectra of B1 and B2, two weak Ce^3+^ in the range of 879–890 eV and 900–910 eV [24] peaks for B2 can be seen but none for B1, which is consistent with the ICP-AES results. Therefore, the impurities in the B1 and B3 are Sn^2+^, while the B2 contains Sn^2+^ and Ce^3+^.

Figure 3 is the XRD patterns of the samples B1–B3. All samples are Cs_2_SnCl_6_ crystal phase with the Fm-3m space group. No impurities’ peaks appeared after the addition of Ce^3+^. Their diffraction peaks shift to smaller angle by comparison with the standard card, as seen from Figure 3b, due to the substitution of Sn^4+^ (r = 0.069 nm) by a larger ion radius of Sn^2+^ (r = 0.112 nm) or Ce^3+^ (r = 0.102 nm). Among three samples, the smallest shift for B2 sample could be attributed to the fact that only B2 has Ce^3+^ from the results of ICP-AES and XPS.

The photoluminescence (PL) spectra of the B1–B3 were plotted in Figure 4. All samples have shown the same broad emission band peaking at 455 nm with a full width at half maximum (FWHM) of 80 nm under 350 nm illumination, according with the broadband emission spectrum peaking at 454 nm in the literature [29]. It suggests that the luminescence of the samples in this work could also originate from a similar defect to the literature [29]. By comparing the defect concentration order, the PL intensity gradually contrarily decreases in the order of B2, B3 and B1 (Figure 4a), suggesting that the PL intensity decreases as increasing defect concentration. The higher defect concentration is easier to form defect clusters, resulting in serious self-absorption, which weakens the luminescence [29]. Therefore, the oxidization of Sn^2+^ in the precursor and crystal to Sn^4+^ possibly causes the concentration of defects decreases as increasing the reaction time. To prove the oxidation process, the new synthesized SnCl_2_ solution was employed as the precursor to synthesize the A-series samples without Ce^3+^ doping. Their XRD patterns were shown in Figure 5. The samples with the reaction time below 60 min composed of Cs_2_SnCl_4_ (PDF#28-0346) and CsSnCl_3_ (PDF#71-2023). Besides, the diffraction peaks of CsSnCl_3_ (PDF#71-2023) crystal existed in 360 min sample. Pure phase Cs_2_SnCl_6_ crystals were obtained as the reaction time prolonged to 660 and 1020 min. A larger amount of Sn^2+^ existed in the new synthesized SnCl_2_ solution since the amount of Sn^2+^ oxidized to Sn^4+^ is relatively smaller, so that the Cs_2_SnCl_6_ crystals together with Cs_2_SnCl_4_ and CsSnCl_3_ were formed at the beginning of the reaction. However, Cs_2_SnCl_4_ and CsSnCl_3_ were gradually oxidized to Cs_2_SnCl_6_ as the reaction time prolonged and finally pure phase Cs_2_SnCl_6_ crystals were obtained.

The crystal structure schematic diagrams of the doped Cs_2_SnCl_6_ were shown in Figure 6. Although the B-series samples contain different types of impurities, their PL and absorption spectra are consistent in peak position and shape, indicating the same type of defects, that is, the composite defects formed by the substitution of Sn^4+^ and Cl vacancies, namely the sample B2 contains [$Ce^{3+}{}_{Sn^{4+}}^{\;}$+V_Cl_] and [$Sn_{Sn^{4+}}^{2+}$+V_Cl_] and only [$Sn_{Sn^{4+}}^{2+}$+V_Cl_] exists in B1 and B3.

Figure 7 is the PL spectra of Ce-doped samples at different concentrations. The undoped sample C1 show very low PL intensity. The intensity was enhanced with increasing the doping concentration and the C3 sample containing Ce^3+^ (listed in Table 1) had the highest intensity. The measured fluorescence quantum efficiency of the C3 sample was 6.54% and the blue light can be observed under ultraviolet light irradiation (shown in the inset of Figure 7). Although the efficiency is much lower than that of Bi^3+^ doped samples [29], as our best knowledge, this is the first report about the PL of the Cs_2_SnCl_6_ crystal.

The PL decay curve of the sample C3 was measured and fitted by the Equation (1), as shown in Figure 8.
(1)I = A exp (−t/τ)

*I* represents the PL intensity of the sample as a function of time, *A* is a constant, *t* is time and *τ* is the fluorescence lifetime of the sample. The fluorescence lifetime *τ* of the C3 sample is 344.6 ns. This is much higher than that of Ce^3+^, such as 25 ns for BaBPO_5_:Ce, 70 ns for Y_3_A1_5_O_12_:Ce [33,34]. It is also much lower than the fluorescence lifetime (5 µs) of the Sn^2+^ dopant luminescent material [35]. These suggested that the PL of the Cs_2_SnCl_6_ crystals in this paper is not derived from Sn^2+^ and Ce^3+^ themselves but the defects formed by them.

The XRD patterns shown in Figure 9 show that the shift degree of the diffraction peaks decreases with the order of C1, C2, C4 and C5. Meanwhile, comparing with the absorption spectra shown in Figure 10, the intensity of the additional weak absorption band decreased gradually. Therefore, it inferred that the concentration of the defects formed by Sn^2+^ reduced gradually because these samples only contained the Sn^2+^ impurity. Although Ce^3+^ was not incorporated into all samples as expected, the addition of Ce^3+^ could impair the content of Sn^2+^ in the crystals as well as the PL intensity. It is presumed that the probability of Sn^2+^ entering the crystal lattice could be reduced with increasing the introduction amount of Ce^3+^, so that the defects caused by Sn^2+^ in the crystals have been reduced. When Ce^3+^ entered the crystal lattice, the total number of defects has been reduced, leading to the PL enhancement within a certain concentration range of the defects. However, the reason of Sn^2+^ defects affected by Ce^3+^ need to be further investigated. Furthermore, only the C3 sample contained Ce^3+^, suggesting the difficulty to introduce Ce^3+^ in this crystal, which can be related to ionic radius and electronegativity [36,37]. The ionic radius of Ce^3+^ (r = 0.102 nm) is quite different from that of Sn^4+^(r = 0.069 nm) and the electronegativity (1.7) of Sn^2+^ is closer to 1.9 of Sn^4+^ [38], compared with the Ce^3+^ (1.2). Therefore, Sn^2+^ could be more likely to replace Sn^4+^ into the crystal. Another reason is that the Sn^2+^ was used as the precursor to enter the crystal together with Sn^4+^.

## 4. Conclusions

The SnCl_2_ solution was used as the precursor and Ce^3+^ was doped to synthesize Cs_2_SnCl_6_ crystals. The undoped and Ce^3+^-doped Cs_2_SnCl_6_ crystals have similar emission spectra peaking at 455 nm with a FWHM of 80 nm. By doping Ce^3+^, the PL intensity was enhanced remarkably. The luminescence of the crystals is derived from the defects introduced by Sn^2+^. When only Sn^2+^ impurities are present in the crystal, [$Sn_{Sn^{4+}}^{2+}$+V_Cl_] defects are formed. In the range of a certain number of defects, the crystal PL intensity increases with the number of defects. When the crystals contain Sn^2+^ and Ce^3+^ impurities, in which the [$Sn_{Sn^{4+}}^{2+}$+V_Cl_] and [$Ce^{3+}{}_{Sn^{4+}}^{\;}$+V_Cl_] defects are formed, the emission of the crystals is the strongest and the fluorescence quantum efficiency is 6.54% with fluorescence lifetime 344.6 ns. 

## Figures and Tables

**Figure 1 materials-12-01501-f001:**
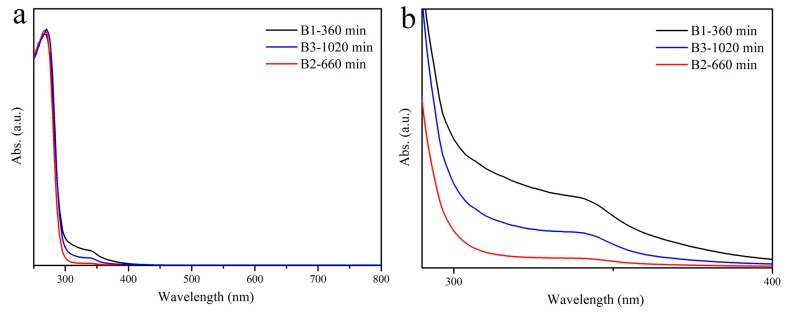
(**a**) absorption spectrum of Cs_2_SnCl_6_ crystal doped with Ce at different reaction times; (**b**) samples of B1, B2, B3 Magnified absorption spectrum of 300–400 nm.

**Figure 2 materials-12-01501-f002:**
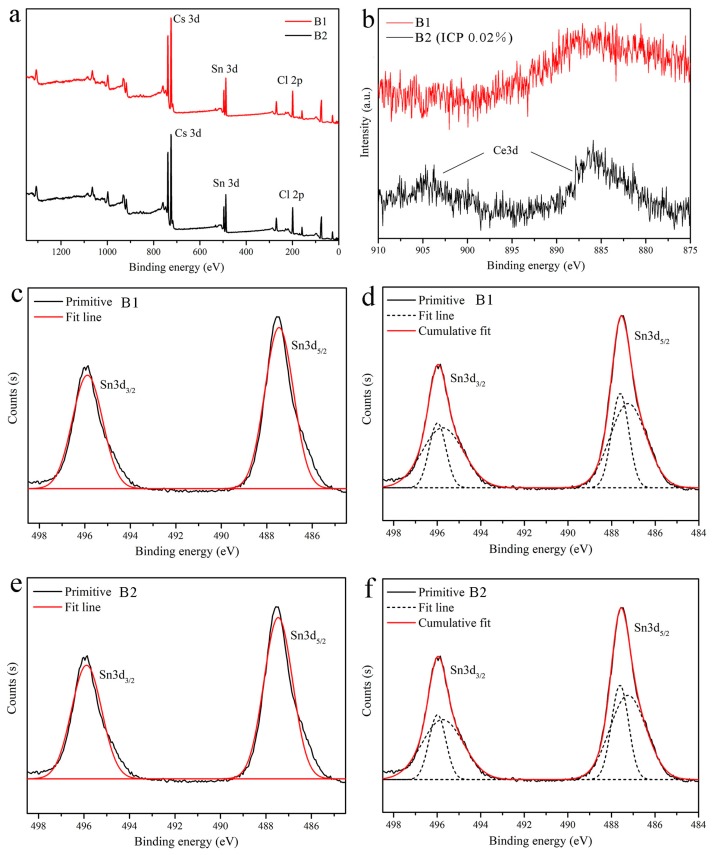
(**a**) XPS survey spectra of the samples B1and B2, (**b**) High-resolution XPS spectra for Ce^3+^ of the samples B1 and B2. (**c**,**d**) High-resolution XPS spectra and peak fitting for Sn 3d of the sample B1. (**e**,**f**) High-resolution XPS spectra and peak fitting for Sn 3d of the sample B2.

**Figure 3 materials-12-01501-f003:**
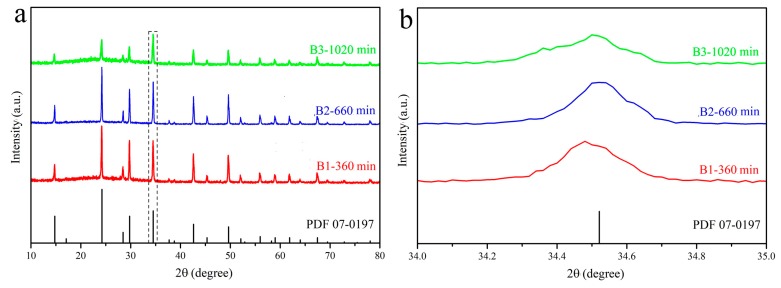
(**a**) The XRD patterns of Ce doped Cs_2_SnCl_6_ prepared at different reaction time; (**b**) the zoom in view in the range of 34–35°.

**Figure 4 materials-12-01501-f004:**
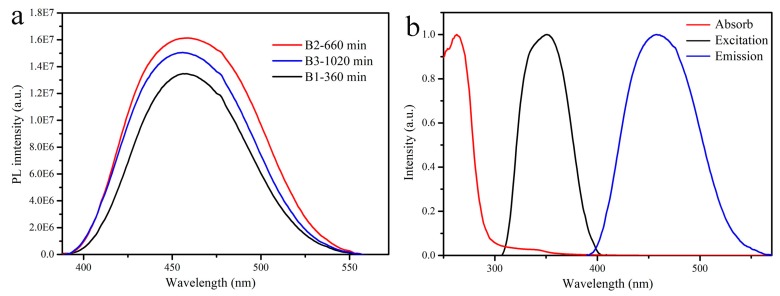
(**a**) The photoluminescence spectra of Ce doped Cs_2_SnCl_6_ crystals at different reaction time; (**b**) the absorption, excitation and emission spectrum of the sample B2.

**Figure 5 materials-12-01501-f005:**
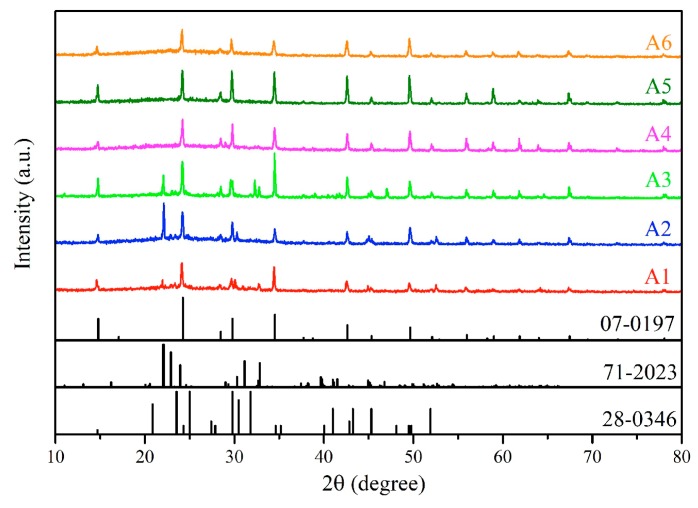
The XRD patterns of Cs_2_SnCl_6_ crystals prepared with newly synthesized SnCl_2_ at different time.

**Figure 6 materials-12-01501-f006:**
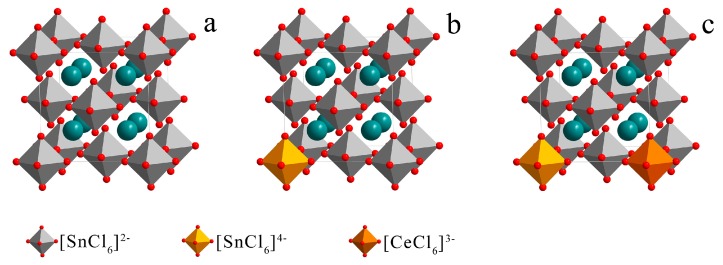
(**a**) The Cs_2_SnCl_6_ crystal structure; (**b**) The Cs_2_SnCl_6_ crystal structure containing [SnCl_6_]^2−^; (**c**) The Cs_2_SnCl_6_ crystal structure containing [SnCl_6_]^4−^ and [CeCl_6_]^3−^.

**Figure 7 materials-12-01501-f007:**
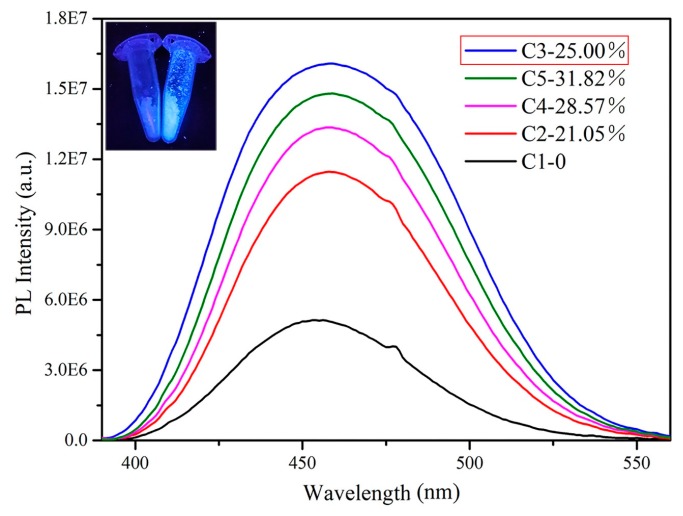
The PL spectra of Cs_2_SnCl_6_ crystals with different Ce doping concentrations, the picture of sample C1 and C3 under ultraviolet light in the illustration.

**Figure 8 materials-12-01501-f008:**
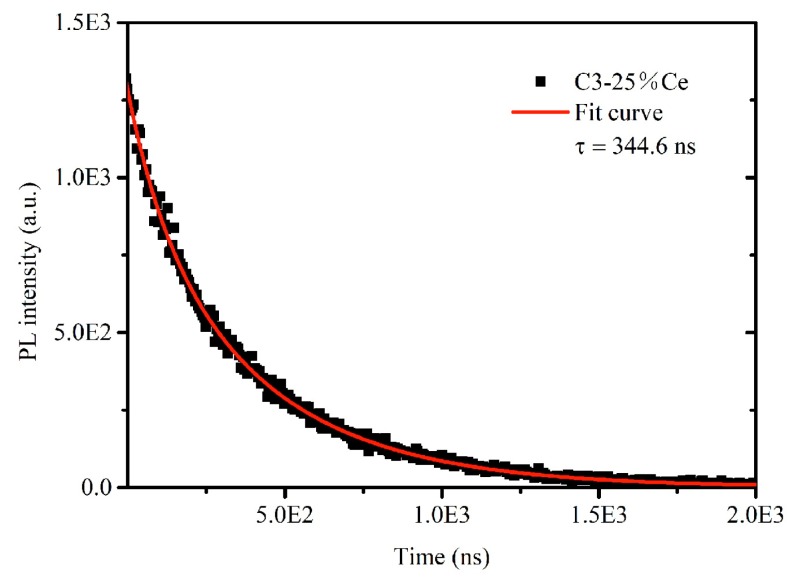
The PL decay curve of the sample C3 and its fit curve.

**Figure 9 materials-12-01501-f009:**
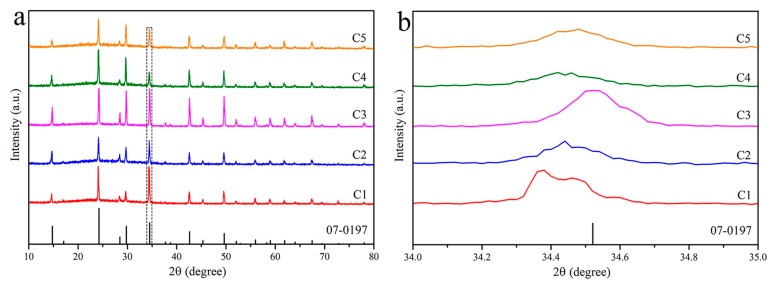
(**a**) The XRD patterns of Cs_2_SnCl_6_ crystals doped with different concentration of Ce; (**b**) the zoom in view in the range of 34–35°.

**Figure 10 materials-12-01501-f010:**
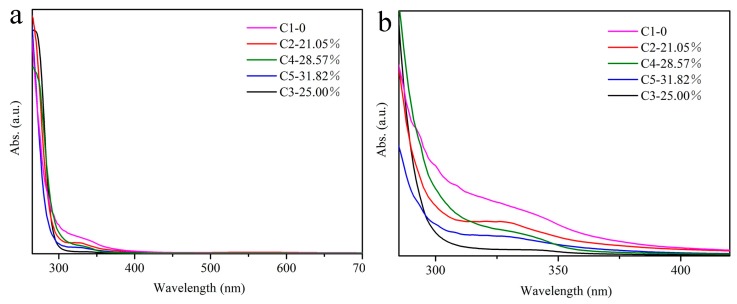
(**a**) The absorption spectra of the samples C1–C5; (**b**) The zoom in view in the range of 300–400 nm.

**Table 1 materials-12-01501-t001:** The synthesis condition and inductively coupled plasma (ICP) data of the samples.

Sample No.	Sn (mol%)	Ce (mol%)	Time (min)	Content (Ce/(Ce + Sn))
A1	100	0	0	–
A2	100	0	10	–
A3	100	0	60	–
A4	100	0	360	–
A5	100	0	660	–
A6	100	0	1020	–
B1	75.00	25.00	360	–
B2	75.00	25.00	660	0.02%
B3	75.00	25.00	1020	–
C1	100	0	660	–
C2	78.95	21.05	660	–
C3(B2)	75.00	25.00	660	0.02%
C4	71.43	28.57	660	–
C5	68.18	31.82	660	–

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
