# Peer review of "Photoluminescence Characteristics of Sn2+ and Ce3+-Doped Cs2SnCl6 Double-Perovskite Crystals"

_materials, 2019, doi:10.3390/ma12091501_

Round 1
Reviewer 1 Report
In this manuscript Yuan et al. report a low toxic and stable Cs2SnCl6 double perovskite crystals were prepared by aqueous phase precipitation method using SnCl2 as precursor. The target material was characterized by means of XRD, ICP-AES, XPS, PL and UV-vis techniques. The manuscript reads well. However, I have made some suggestions and corrections before its acceptance, please see them in the attached file.

Author Response
I am sincerely thank the reviewer for his comments, I revised all mentioned error in format and grammar, as be seen in the revised manuscript.
Reviewer 2 Report
Photoluminescence characteristics of Sn2+ and Ce3+ Doped Cs2SnCl6 double-perovskite Crystals 3
Hongdan Zhang, Ludan Zhu, Jun Cheng, Long Chen, Chuanqi Liu and Shuanglong Yuan
The work reported by Zhang et al is experimentally conducted. It involves synthesis and measurements of various properties using techniques such as X-ray photoelectron spectroscopy and X-ray diffractometry and a few others. The study has been carefully conducted to show that all-inorganic Cs2SnCl6 double perovskite crystals are stable and are better than similar other materials. Although the research can be interesting to the perovskite audience, there are several issues that are necessary to be answered. This is essential for the improvement of the ms.
1 – The introduction section needs improvement, especially the writing part.
2 – There are many long sentences scattered all over the ms. This causes difficulty in understanding. It does enable me to understand what the authors have tried to convince. Thus, I suggest a careful and thorough revision.
3 – There are grammatical mistakes all over the ms. I did not see any smoothness between lines.
4 – The authors wrote that “the C3 sample contains Ce3+, suggesting the difficulty to introduce Ce3+ in this crystal, which could be related to ionic radius and electronegativity”.
Comment: Is this conclusion a speculation? Is it a problem with the ion radius and electronegativity of the ions? Or are there other factors that need to be considered to resolve the anomaly?
5 – I cannot see the crystallographic information file, as well as the details of the structural data. Can the authors provide them at least for review?
6 – There is no rigorous discussion about the bandgap of the material. It is a very important property of any perovskites (hybrid, metal-organic or all-inorganic!). Can this part be elaborated in a detail? The authors have shown some peak absorptions using spectral measurements. This is not the entire story of any energy materials. Interpretation and appropriate citation of previous literature are essential.
7 – Why did the authors conclude that the materials examined are stable compared to any other related perovskites? Is it because of Sn and Ce or something else? This should be clarified.
8 – Many background references are missing in this paper. Such papers much be cited during the revision.
Author Response
Point 1:The introduction section needs improvement, especially the writing part.
Response 1: Thanks to the comments, we revised the introduction section, the revised parts have been figured out with “track changes”model.
Point 2:There are many long sentences scattered all over the ms. This causes difficulty in understanding. It does enable me to understand what the authors have tried to convince. Thus, I suggest a careful and thorough revision.
Response 2: Sincerely thanks to the review for the comment, some long sentences have been rewritten for easy understanding, these changes also can be seen in the revised manuscript.
Point 3: There are grammatical mistakes all over the ms. I did not see any smoothness between lines.
Response 3: Sorry for many grammatical mistakes, we tried my best to revise these mistakes and improve the sentences, these changes have also been figured out in the revised manuscript.
Point 4: The authors wrote that “the C3 sample contains Ce3+, suggesting the difficulty to introduce Ce3+ in this crystal, which could be related to ionic radius and electronegativity”.
Comment: Is this conclusion a speculation? Is it a problem with the ion radius and electronegativity of the ions? Or are there other factors that need to be considered to resolve the anomaly?
Response 4: Thanks for the review’s comment, the substitution of doped ions for main lattice ions is related to ion radius and electronegativity difference according to the literature[36-37], therefore, large ion radius and electronegativity difference between Ce3+and Sn4+makes the substitution difficult. Of course, the synthesis condition can affect the degree of substitution. For this work, too long or too short reaction time leaded to the difficulty of Ce3+substitution. This means it needs proper synthesis condition. This is a clear conclusion so we changed “could” to “can”.
Point 5: I cannot see the crystallographic information file, as well as the details of the structural data. Can the authors provide them at least for review?
Response 5: we will attach the CIF file together with the submission of the revised version.
Point 6: There is no rigorous discussion about the bandgap of the material. It is a very important property of any perovskites (hybrid, metal-organic or all- inorganic!). Can this part be elaborated in a detail? The authors have shown some peak absorptions using spectral measurements. This is not the entire story of any energy materials. Interpretation and appropriate citation of previous literature are essential.
Response 6: As the review mentioned, the bandgap of the materials is very important. The bandgaps of Cs2SnCl6and Bi-doped one have been calculated and investigated in detail in the literature [29](Tan, Z.; Li, J.; Zhang, C.; Li, Z.; Hu, Q.; Xiao, Z.; Kamiya, T.; Hosono, H.; Niu, G.; Lifshitz, E.; Cheng, Y. Tang, J. Highly Efficient Blue-Emitting Bi-Doped Cs2SnCl6 Perovskite Variant: Photoluminescence Induced by Impurity Doping. Adv. Funct. Mater. 2018, 28, 1801131.). Because of the similar characteristic of the absorption spectra, which means the similar bandgap structure, we cited their conclusion in this work.
Point 7: Why did the authors conclude that the materials examined are stable compared to any other related perovskites? Is it because of Sn and Ce or something else? This should be clarified.
Response 7: For typical ABX3perovskites, the stability is a key problem indeed. This is one of the reasons why we studied the PL of Cs2SnCl6crystals. The high stability is attributed to the high value Sn4+and higher decomposition enthalpy (1.37 eV per formula unit for Cs2SnCl6) than that of traditional ABX3 halide perovskites (e.g., 0.29 eV for CsPbBr3) [29]. Thus, we added the above sentence to the introduction part. Actually, we have also investigated their stability in water for different time and in the air at different temperature for 30 min. It exhibited remarkable stability in different environment conditions compared with typical ABX3perovskites without surface coating. However, these results have not been shown in this manuscript due to too long for communication
Point 8: Many background references are missing in this paper. Such papers much be cited during the revision.
Response 8:Thanks for the comment. We added some necessary references in the introduction part as seen in the revised manuscript.
Round 2
Reviewer 2 Report
I am happy to see the authors have revised the ms based on my comments. I can now recommend publication of this paper in materials. However, I still ask the authors to revise their ms seriously to make sure that all kinds of errors are elimnated.